# Ibudilast Mitigates Delayed Bone Healing Caused by Lipopolysaccharide by Altering Osteoblast and Osteoclast Activity

**DOI:** 10.3390/ijms22031169

**Published:** 2021-01-25

**Authors:** Yuhan Chang, Chih-Chien Hu, Ying-Yu Wu, Steve W. N. Ueng, Chih-Hsiang Chang, Mei-Feng Chen

**Affiliations:** 1Bone and Joint Research Center, Chang Gung Memorial Hospital, No. 5, Fuxing Street, Guishan Dist., Taoyuan 33305, Taiwan; yhchang@cgmh.org.tw (Y.C.); r52906154@cgmh.org.tw (C.-C.H.); l43219713@hotmail.com (Y.-Y.W.); wenneng@cgmh.org.tw (S.W.N.U.); 2Department of Orthopedic Surgery, Chang Gung Memorial Hospital, Taoyuan 33305, Taiwan; 3College of Medicine, Chang Gung University, Taoyuan 33302, Taiwan; 4Graduate Institute of Clinical Medical Sciences, College of Medicine, Chang Gung University, Taoyuan 33302, Taiwan

**Keywords:** lipopolysaccharide, osteoblast, osteoclast, bone healing, bone bridge, callus bone, femoral defect, ibudilast

## Abstract

Bacterial infection in orthopedic surgery is challenging because cell wall components released after bactericidal treatment can alter osteoblast and osteoclast activity and impair fracture stability. However, the precise effects and mechanisms whereby cell wall components impair bone healing are unclear. In this study, we characterized the effects of lipopolysaccharide (LPS) on bone healing and osteoclast and osteoblast activity in vitro and in vivo and evaluated the effects of ibudilast, an antagonist of toll-like receptor 4 (TLR4), on LPS-induced changes. In particular, micro-computed tomography was used to reconstruct femoral morphology and analyze callus bone content in a femoral defect mouse model. In the sham-treated group, significant bone bridge and cancellous bone formation were observed after surgery, however, LPS treatment delayed bone bridge and cancellous bone formation. LPS inhibited osteogenic factor-induced MC3T3-E1 cell differentiation, alkaline phosphatase (ALP) levels, calcium deposition, and osteopontin secretion and increased the activity of osteoclast-associated molecules, including cathepsin K and tartrate-resistant acid phosphatase in vitro. Finally, ibudilast blocked the LPS-induced inhibition of osteoblast activation and activation of osteoclast in vitro and attenuated LPS-induced delayed callus bone formation in vivo. Our results provide a basis for the development of a novel strategy for the treatment of bone infection.

## 1. Introduction

Bacterial infection presents a challenge in orthopedic surgery, requiring effective antibiotic treatment, debridement of infected tissues, and maintenance of fracture stability. There is some evidence that antibiotic-killed bacteria may still exert a negative effect on bone healing or remodeling. However, only a few animal model-based studies have investigated this issue. The post-infectious inflammatory response elicited by dead bacteria prolongs bone destruction, despite antibiotic treatment in vivo [1]. Another study showed that metals with bacterial attachment prolong fracture healing [2]. Additionally, simultaneous treatment with systemic immunosuppressive agents and antibiotics improves outcomes in a mouse model of bacterial arthritis and sepsis [3]. These studies suggest that antibiotic-killed bacteria still impair skeletal function. The relevant literature implies that dead bacteria mainly impair bone healing via bacterial cell wall components such as lipopolysaccharides (LPS), which trigger immune responses via toll-like receptor 4 (TLR4). Gram-negative bacteria contain LPS in their cell walls. The systemically circulating endotoxin may derived from antibiotic-killed bacteria, minor infections, intestinal flora, or dental procedures and may bind to wear particles [4,5,6].

In patients who have received an artificial joint replacement or other orthopedic metal implants, endotoxin causes aseptic implant loosening. Endotoxin-adherent wear particles may contribute to the aseptic loosening of orthopedic implants, even in the absence of clinical or microbiological evidence of infection [4,7,8]. Tatro et al. indicated that endotoxin wear particles on orthopedic implants may regulate the rate of osteolysis in a murine calvaria model as well as in patients with aseptic loosening [5]. Tomomatsu et al. demonstrated that the bone mineral density of the tooth socket is significantly reduced by LPS injection after tooth extraction in a murine model [9]. Using LPS-doped polyethylene particles, Liu et al. demonstrated that the particle-induced impairment of fixation was directly associated with increased bone resorption and depressed bone formation in a rat model, supporting the clinical phenomenon of particle-related implant osteolysis and loosening [10]. In LPS-injected rats, an exaggerated inflammatory environment may reduce the bone morphogenetic protein-2 (BMP-2)/absorbable collagen sponge-induced bone mass by suppressing BMP-2-induced osteoblastic differentiation and increasing the number or activity of osteoclasts [11]. Abu-Ame et al. confirmed that LPS-induced osteoclastogenesis is mediated by TNF (tumor necrosis factor) and/or its p55 receptor in vivo [12]. Despite evidence that LPS inhibits osteoblast differentiation, the effect of LPS on bone healing is unclear. Therefore, the main goal of this study was to investigate the effect of LPS on the bone healing process and to determine whether ibudilast, a TLR4 antagonist, has beneficial effects on LPS-impaired bone healing in vivo.

## 2. Results

### 2.1. LPS Delayed Bone Healing in Mice with Femoral Bone Defects

Because we want to analyze the effects of LPS on bone healing in femoral defect mouse model. We administered two doses of LPS to mice after bone defect surgery, including 2 mg/kg (low dose) or 10 mg/kg (high dose). Micro-computed tomography (microCT) images were used to reconstruct the femoral morphology and analyze the callus bone content (Figure 1A). In the sham-treated group of femoral-defect mice, significant bone bridge and cancellous bone formation were observed on day 14 after surgery (Figure 1B,C). Low-dose LPS treatment resulted in delayed bone bridge and cancellous bone formation, moreover, high-dose LPS treatment resulted in full fracture. Additionally, low-dose LPS significantly decreased the trabecular number and trabecular bone thickness in the bone bridge area compared with those in the sham-treated group (Figure 1D). These trabecular bone parameters could not be quantified in the high-dose LPS group owing to full fracture. Masson’s trichrome staining showed bone bridge formation in the sham-treated group (Figure 1E). LPS also delayed weight recovery after surgery (Figure 1F). C-terminal telopeptides of type I collagen (CTX1) and osteocalcin concentration are indicators of bone metabolism. LPS significantly increased CTX1 and osteocalcin concentrations in the serum (Figure 1G). 

### 2.2. LPS had Adverse Effects on Bone Formation In Vitro

LPS directly significantly inhibited osteogenic factor-induced MC3T3-E1 cell differentiation, including the alkaline phosphatase (ALP) level (Figure 2A,B) and calcium deposition (Figure 2C,D). LPS also decreased osteopontin secretion during MC3T3-E1 cell differentiation (Figure 2E). LPS also directly and significantly increased the expression level of osteoclast-associated molecules, including cathepsin K (Figure 3A–C) and tartrate-resistant acid phosphatase (TRAP; Figure 3D,E). 

### 2.3. Effects of Ibudilast on LPS-Induced Inhibition of Bone Formation In Vitro

We next examined whether ibudilast, a TLR4 antagonist, can reverse the adverse effects of LPS on bone formation in vitro. Both staining and quantitative results revealed that ibudilast can effectively reverse the LPS-induced effects on ALP levels and calcium deposition in MC3T3-E1 cells (Figure 4A–C). Moreover, ibudilast significantly inhibited the LPS-induced increase in TRAP-positive osteoclasts (Figure 4D,E). The results of these in vitro analyses indicate that ibudilast inhibits the adverse effects of LPS on bone formation.

### 2.4. Ibudilast Mitigates LPS-Delayed Callus Bone Formation in Mice with Femoral Bone Defects

Animal experiments were subsequently conducted to confirm that ibudilast can protect against delayed bone healing by LPS in vivo (Figure 5A). Micro-CT images of mouse femoral bone revealed that ibudilast administration following LPS treatment resulted in a slightly enhanced callus bone formation (Figure 5B). Moreover, the number of bone trabeculae was higher after ibudilast administration than after treatment with LPS alone (Figure 5C). Masson’s trichrome staining revealed bone bridge formation on day 14 after the bone defect had appeared in control group (Figure 5D). LPS impaired the bone bridge formation. However, in the ibudilast-treatment group, a dense bone bridge appeared after introducing femoral defects. These results demonstrated that ossification associated with bone healing was nearly complete on day 14 in the ibudilast-treated group but not in the LPS-only group.

## 3. Discussion 

### 3.1. Effects of LPS on Osteoblasts

LPS released from the bacterial cell wall leads to the stimulation of TLR4 [13]. Although a few studies have investigated the effect of LPS on osteoblasts and osteoclasts, controversial and inconsistent results have been obtained. LPS suppresses the mRNA expression of Runx2 (runt-related protein 2), osterix (*Sp7*), and activating transcription factor 4 (*ATF4*) during osteoblast differentiation in the wild-type, but not in the myd88(−/−) osteoblasts [14]. Although LPS adherent to titanium alloy discs had no detectable effects on attachment, spreading, nor growth in the early stages of MC3T3-E1 osteogenesis in vitro, osteogenic differentiation and mineralization were inhibited by adherent LPS [15]. Based on these results, bacterial debris can act as a form of surface contaminant that may impair the osseointegration of orthopedic implants [15]. LPS induces apoptosis and inhibits osteoblast differentiation via the JNK (c-Jun NH 2-terminal kinases) pathway in preosteoblastic cells (MC3T3-E1) [16]. An inflammatory environment triggered by LPS in vitro suppressed BMP-2-induced osteogenic differentiation of bone marrow stromal cells (BMSCs), as evidenced by decreased alkaline phosphatase (ALPase) activity and downregulated osteogenic genes [17]. LPS inhibits BMP-2-induced osteogenic differentiation, and the crosstalk between TLR4/MyD88/NF-κB and BMP/Smad signaling negatively modulates the osteoinductive capacity of BMP-2 [17]. LPS stimulates periosteal osteoclast formation and bone resorption by stimulating RANKL (receptor activator of nuclear factor-κB ligand) in osteoblasts [18]. According to these findings, although the detailed mechanism remains unclear, LPS seems to inhibit osteoblastogenesis. In this study, our results demonstrated that LPS delays bone healing in mice with femoral bone defects. Moreover, ibudilast mitigates the LPS-induced delay in callus bone formation in mice with femoral bone defects. 

### 3.2. Effects of LPS on Osteoclasts

TLRs have been implicated in promoting osteoclast-mediated bone resorption that is associated with inflammatory conditions [19,20,21]. LPS completely abrogates the RANKL-induced expression of B lymphocyte-induced maturation protein-1 (Blimp1), a global transcriptional repressor of anti-osteoclastogenic genes encoding Bcl6, IRF8, and MafB [22]. Xiao et al. have shown that in infection-induced inflammation, the components of invading bacteria (e.g., LPS) activate macrophages. Factors secreted from activated macrophages then induce SPHK1 activity in bone marrow stromal cells (BMSCs), thereby resulting in the overproduction of S1P [23]. S1P acts on the S1PR1 receptor S1PR1 in BMSCs, which results in the overproduction of RANKL and eventually induces osteoclastogenesis and bone resorption via the activation of RANK in osteoclast precursors [23]. Thus, the SPHK1–S1PR1–RANKL axis regulates interactions between macrophages and BMSCs in inflammatory bone loss [23]. Here, we used our femoral defect mouse model to confirm that LPS induced increases in osteoclast activation, which could then be abrogated by treatment with ibudilast. This intervention also resulted in bone bridge formation and promoted mature osteocytes formation, thereby demonstrating that our model can reflect the bacterial cell wall induction of bone reduction and therapeutic resolution of this condition.

### 3.3. Innovation of this Study

A disruption in the balance between bone resorption by osteoclasts and bone formation by osteoblasts is observed in bone infection [24]. Bone resorption can be regulated by the differentiation of new osteoclasts and the activation of mature osteoclasts. Bone formation can be modulated by osteoblast differentiation. Our results provide insight into interactions between osteoblasts/osteoclasts and bacterial cell walls. Importantly, our mouse model can be used for further analyses of the contributions of new factors to the interaction between osteoblasts and osteoclasts to develop new therapeutic targets for bone healing. We believe that our research findings will facilitate the development of a novel strategy for the treatment of bone infection and will contribute to the field of bone physiology. 

## 4. Materials and Methods

### 4.1. Experimental Animal Study

All animal procedures complied with the National Institute of Health guidelines and were reviewed and approved by the local Hospital Animal Care and Use Committee. All procedures were approved by the Chang Gung Memorial Hospital review committee (IACUC approval number 2018121701). Ten-week-old male C57BL/6NCrlBltw mice were used for the femoral defect model [25,26]. 

Initially, mice were anesthetized via intraperitoneal injection (0.01 mL/kg body weight) of a 1:1 (*v*/*v*) mixture of tiletamine-zolazepam (Zoletil, Virbac, Carros, France) and xylazine hydrochloride (Bayer HealthCare AG, Leverkusen, Germany), and the surgical site was shaved and disinfected with povidone-iodine. An incision was made in the skin overlying the right knee joint. A medial parapatellar arthrotomy (with lateral displacement of the quadriceps–patella complex) was performed to access the distal femur. After locating the femoral intercondylar notch, the femoral intramedullary canal was manually pierced with a 25-gauge needle and intrafemorally injected with 2 or 10 mg/kg LPS (from *Escherichia coli* O127:B8; Sigma-Aldrich, St. Louis, MO, USA) in phosphate-buffered saline (PBS; 10 μL). A stainless-steel rod (with a length of 0.9 mm and diameter of 0.4 mm) was surgically placed in a retrograde manner. A 1-mm defect was formed at the midshaft of the right femur using drill bits of different sizes. The quadriceps–patellar complex was repositioned to the midline, and the surgical site was closed with subcutaneous 6-0 Dexon sutures. Finally, the painkiller buprenorphine (0.2 mg/kg) was administered subcutaneously every 24 h as an analgesic throughout the experimental duration. The mice were sacrificed on day 7 or 14 post-surgery (5–8 mice per group). The femur was immediately fixed in formaldehyde (10%) and subjected to micro-CT analysis.

### 4.2. Serum Osteocalcin and CTX1 Assay

The serum was collected on Day 14 post-surgery. Serum osteocalcin and CTX1 assay were measured using commercially available ELISA kits (osteocalcin, BioSource Europe SA, Nivelles, Belgium; CTX1, San Diego, CA, USA), according to the manufacturers’ protocols.

### 4.3. Micro-CT Bone Imaging

A nondestructive ultrastructural analysis was performed using a SkyScan 1176 micro-CT scanner (Bruker MicroCT, Kontich, Belgium). The samples were scanned at 50 kV with a 0.5-mm aluminum filter. Images with a resolution of 9 μm were reconstructed using GUP-NRecon software (version 1.7.4.2) and analyzed using CTAn software (version 1.15.4.0, SkyScan). The grayscale was based on Hounsfield units, and validated calcium standards were scanned as a density reference (0.25 and 0.75 g/cm^3^ Hydroxyapatite Phantoms). The trabecular number, trabecular thickness and trabecular separation were assay. A 3D image was constructed using CTVox software (version 3.3.0, SkyScan) for illustration purposes. For the detailed procedure, please refer to our previously published studies [25,26].

### 4.4. Histochemistry and Immunofluorescence Staining

Femur samples were decalcified and embedded in paraffin. Four-micrometer-thick sections were stained with hematoxylin and eosin or Masson’s trichrome stain. Slides were digitalized using a NanoZoomer S360 digital slide scanner (Hamamatsu Photonics, Hamamatsu, Japan). Each image was acquired under a microscope (DFC7000 T; Leica Microsystems, Wetzlar, Germany).

### 4.5. Osteoblast and Osteoclast Differentiation

MC3T3-E1 cells (CRL-2593; American Type Culture Collection, Manassas, VA, USA) and RAW264.7 cells were maintained in alpha-modified Eagle’s medium supplemented with 10% fetal bovine serum (FBS) and antibiotics. For osteoclast differentiation, MC3T3-E1 cells were cultured in 10% FBS-differentiation medium with or without osteogenic factors (OS), which were composed of 5 mM glycerol 2-phosphate, 0.1 µM dexamethasone, and 50 mM ascorbic acid. Cells were treated with or without 100 ng/mL LTA in osteogenic differentiation medium. Calcification of MC3T3-E1 cells was assessed by Alizarin Red S staining (ScienCell, Carlsbad, CA, USA) to monitor matrix mineralization. Calcium assays were performed using a Calcium LiquiColor Assay (Stanbio Laboratory, Boerne, TX, USA) in accordance with the manufacturer’s instructions. A Fast Violet B Salt capsule (catalog number 851–10 CAP; Sigma-Aldrich) was dissolved in naphthol AS-MX phosphate alkaline solution (catalog number 855; Sigma-Aldrich) and then used for ALP staining. Sigma 104^®^ phosphatase substrate (catalog number 104105; Sigma-Aldrich; St. Louis, MO, USA) was used for ALP assays.

For osteoclast differentiation, RAW264.7 cells were differentiated for five days into mature osteoclasts in the presence of LPS (100 ng/mL) or vehicle solution alone. The formation of mature osteoclasts was assessed by immunostaining for cathepsin K (ab19027; Abcam) and TRAP (Takara, Shiga, Japan). F-actin was stained with Alexa Fluor 647-conjugated phalloidin (A22287; Invitrogen, Carlsbad, CA, USA), and DAPI (4′,6-diamidino-2-phenylindole) staining was used for nuclear staining (D1306; Invitrogen; Waltham, MA, USA).

### 4.6. RNA Extraction and Quantitative Polymerase Chain Reaction

Total RNA was extracted from cell pellets using the RNeasy Mini Kit (Qiagen, Valencia, CA, USA) in accordance with the manufacturer’s protocol. Quantitative polymerase chain reaction (qPCR) was performed using iQ SYBR Green Supermix (Bio-Rad Laboratories, Hercules, CA, USA) with specimens and diluted standards. Serial dilutions of purified DNA were used for qPCR calibration. PCR-grade water was used as a negative control. The primer sequences used for amplification were as follows: mouse glyceraldehyde 3-phosphate dehydrogenase (*GAPDH*), 5′-AATGGTGAAGGTCGGTGTG-3′ (forward) and 5′-GTGGAGTCATACTGGAACATGTAG-3′ (reverse); mouse cathepsin K (*CTSK*), 5′-AGGGAAGCAAGCACTGGATA-3′ (forward) and 5′-GCTGGCTGGAATCACATCTT-3′ (reverse). Experiments were performed in triplicate, with coefficients of variation < 5%. *CTSK* expression was normalized against *GAPDH* expression. 

### 4.7. Ibudilast Treatment

Following the intrafemoral delivery of LPS, mice received daily intraperitoneal injections 4 mg/kg/day of either ibudilast (Cayman Chemical, Ann Arbor, MI, USA) or a vehicle (PBS) for 3 consecutive days. For the in vitro effects of ibudilast (10 µM), cells were cultured in the presence of ibudilast or vehicle solution alone.

### 4.8. Statistical Analysis

All data were obtained from at least three independent experiments. Quantitative data were analyzed by two-way analysis of variance (ANOVA), followed by Bonferroni’s post hoc tests. Results are presented as means ± standard error of the mean. Body weights were analyzed by two-way repeated-measures ANOVA, followed by Tukey’s post-hoc tests. Categorical variables were examined using GraphPad Prism, version 7.0 (GraphPad Inc., San Diego, CA, USA). Two-tailed *p*-values of < 0.05 were considered to reflect statistically significant differences.

## Figures and Tables

**Figure 1 ijms-22-01169-f001:**
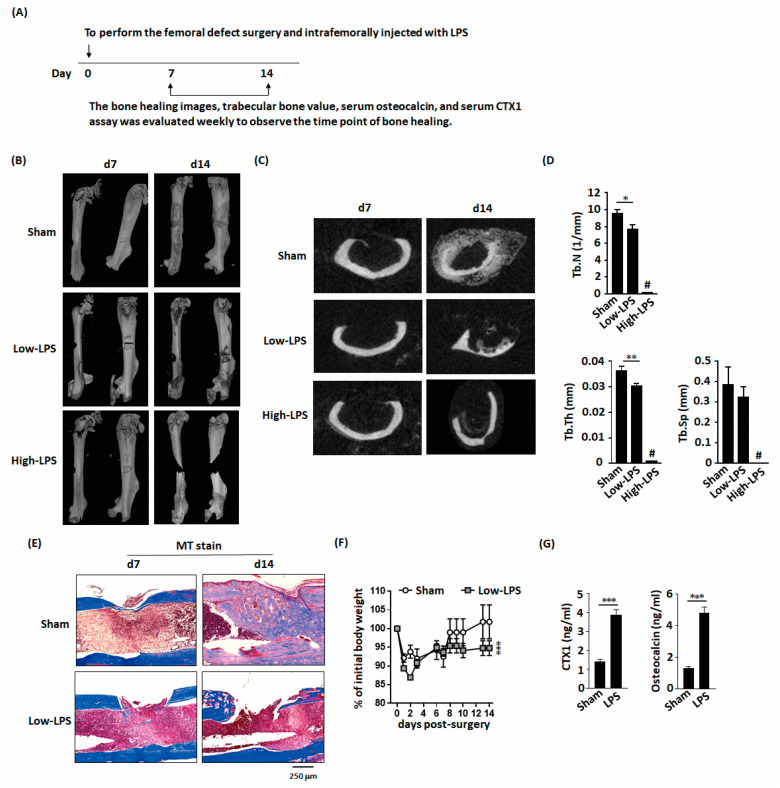
Lipopolysaccharides (LPS) delayed new bone formation and bone healing in mice with femoral bone defects. (**A**) Schematic representation of the experimental setup. (**B**) Micro-computed tomography (microCT) three-dimensional image of the bone defect site show that low-dose LPS delays bone healing and high-dose LPS promotes bone fracture in the bone defect area. (**C**) Representative transverse images of the fractured femur by microCT. (**D**) LPS decreases the trabecular number and thickness. (**E**) MT stained histological sections demonstrate LPS inhibits bone bridge formation during bone healing. (**F**) LPS impairs body weight recovery. (**G**) LPS increases the serum levels of CTX1 and osteocalcin, an indicator of the bone turnover rate. Data are presented as the mean ± standard error of the mean. Analyses were conducted with two-way analysis of variance followed by Bonferroni’s post-hoc test. ** p* < 0.05, *** p* < 0.01, *** *p* < 0.001. Abbreviations: LPS, lipopolysaccharide; Tb.N, trabecular number; Tb.Th, trabecular thickness; Tb.Sp, trabecular separation; MT, Masson’s trichrome stain; CTX1, C-terminal telopeptides of type I collagen; #, full fracture. Sham, *n* = 5; Low-LPS, *n* = 7; and High-LPS, *n* = 5.

**Figure 2 ijms-22-01169-f002:**
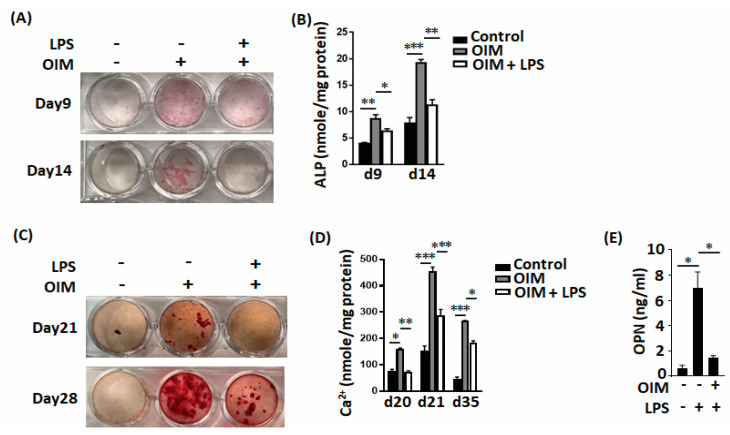
LPS inhibits osteoblastic cell differentiation in vitro. (**A**,**B**) LPS inhibits both ALP (alkaline phosphatase) staining and ALP concentrations during osteoblast differentiation. (**C**,**D**) LPS inhibits calcium deposits during osteoblast differentiation. (**E**) LPS decreased OPN production during osteoblast differentiation. Data are presented as the mean ± standard error of the mean. Analyses were conducted with two-way analysis of variance followed by Bonferroni’s post-hoc test. ** p* < 0.05, *** p* < 0.01, *** *p* < 0.001. Abbreviations: OS, osteogenic factor; ALP, alkaline phosphatase; OPN, osteopontin.

**Figure 3 ijms-22-01169-f003:**
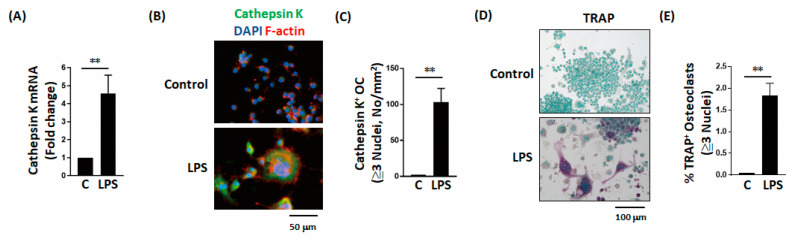
LPS enhances osteoclastic cell differentiation in vitro. (**A**–**C**) LPS not only increases cathepsin K expression but also increases the number of cathepsin K-positive cells during osteoclast differentiation. (**D**,**E**) LPS increases both the expression intensity of TRAP and the percentage of TRAP-positive cells during osteoclast differentiation. Data are presented as the mean ± standard error of the mean. Analyses were conducted with two-way analysis of variance followed by Bonferroni’s post-hoc test. *** p* < 0.01. Abbreviations: TRAP, tartrate-resistant acid phosphatase; LPS, lipopolysaccharides; OC, osteoclasts; C, vehicle control.

**Figure 4 ijms-22-01169-f004:**
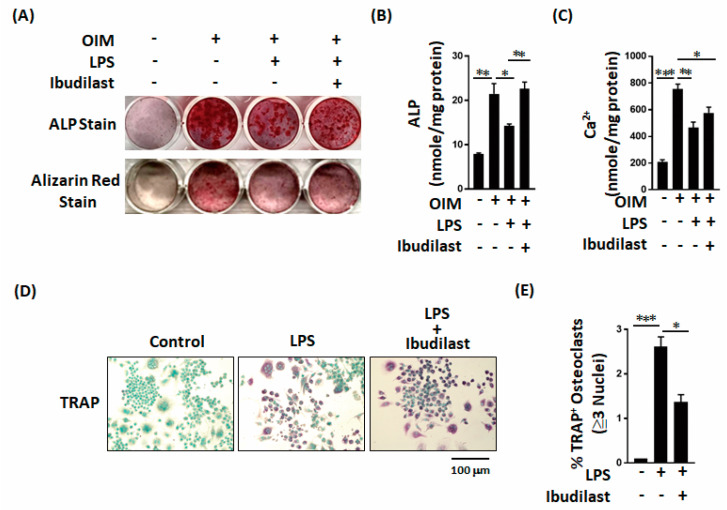
Ibudilast not only reverses the LPS-induced decrease in ALP production but also prevents the LPS-induced increase in osteoclast activation. (**A**–**C**) Ibudilast reversed the LPS-induced inhibition of alkaline phosphatase and bone calcium parameters in vitro. (**D**,**E**) Ibudilast prevented the LPS-induced increase in TRAP-positive osteoclast percentage in vitro. Data are presented as the mean ± standard error of the mean. Analyses were conducted with two-way analysis of variance followed by Bonferroni’s post-hoc test. ** p* < 0.05, *** p* < 0.01, *** *p* < 0.001. Abbreviations: OS, osteogenic factor; ALP, alkaline phosphatase.

**Figure 5 ijms-22-01169-f005:**
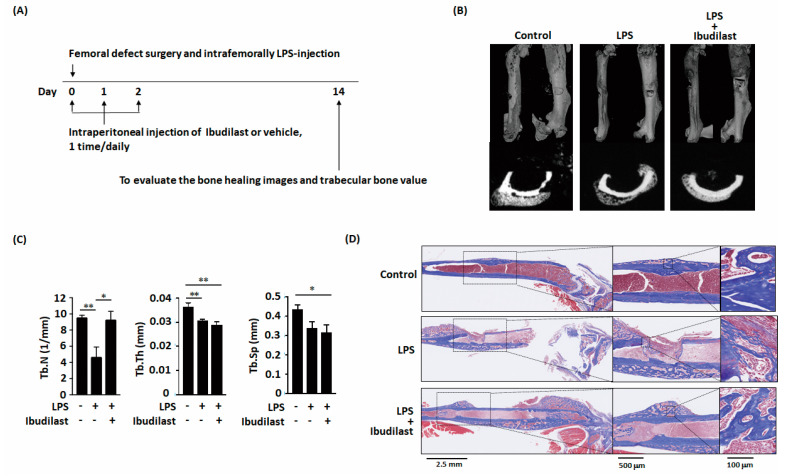
Ibudilast prevents LPS-induced decreases in trabecular number in vivo. (**A**) Schematic representation of the experimental setup. (**B**,**C**) MicroCT three-dimensional image and transverse images of the bone defect site show that Ibudilast reversed the LPS-induced decrease in trabecular number in vivo. (**D**) Ibudilast not only promoted bone bridge formation but also promoted mature osteocyte formation. Data are presented as the mean ± standard error of the mean. Analyses were conducted with two-way analysis of variance followed by Bonferroni’s post-hoc test. ** p* < 0.05, *** p* < 0.01. Abbreviations: LPS, lipopolysaccharide; Tb.N, trabecular number; Tb.Th, trabecular thickness; Tb.Sp, trabecular separation. Sham, *n* = 5; LPS, *n* = 7; and LPS + Ibudilast, *n* = 5.

## Data Availability

Data sharing is not applicable to this article.

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
