# Peer review of "Ibudilast Mitigates Delayed Bone Healing Caused by Lipopolysaccharide by Altering Osteoblast and Osteoclast Activity"

_ijms, 2021, doi:10.3390/ijms22031169_

Round 1

Reviewer 1 Report

In the present work, the authors aimed to contribute to the understanding of the mechanisms elicited by bacterial cell wall components in the framework of bone healing. To this purpose, they performed in vitro and in vivo studies. Specifically, they induced osteogenic differentiation of MC3T3 cells and osteoclast differentiation of RAW264 cells in the presence of LPS [[as compared to? this is not clear: vehicle only? no RANKL? this could be the reason why no osteoclasts are seen in the control condition in Fig3D and Fig4D, and osteoclast marker genes are extremely low. Actually, we do not think this is correct, the condition RAW + RL should be presented, and also RAW + RL + LPS]]. They found that in vitro LPS treatment reduced osteoblast differentiation and function, while increased osteoclastogenesis. Moreover, they observed healing outcome in a mouse model of bone fracture repair in the presence of LPS, and found a delay in bone bridge and cancellous bone formation. These effects were blocked in the presence of a TLR4 antagonist (ibudilast, erroneously indicated as a TLR4 agonist in the abstract; please, modify), pointing to a possible strategy for the treatment of bone infection.

The topic is of interest, but according to this reviewer overall the manuscript presents limited novelty. Large part of the results presented here appear to fall within current knowledge in the field. The authors themselves had already published several observations in a previous recent paper (Chen et al, J Clin Med 2019; and in fact, LTA at line 218, page 7 is probably a leftover from that paper). In our mind, in the framework of the present manuscript, all these data shall constitute background knowledge, not original results. We would suggest the authors modify the manuscript according to our concern and further characterize the cellular and molecular effects of the proposed treatment. 

Other observations:

since the Methods section appears after the Results, this latter should briefly describe the experimental setting before presenting data, to facilitate understanding. For example, in subsection 2.1, what is the low dose and what is the high dose for LPS treatment? how and when is it administered? at which time point is the assessment of CTXI and osteocalcin serum level performed? and so on. 

A similar observation applies to the Figure legends: the presence of a title giving the overall message of the figure is fine, but then the text should describe what is presented in each panel. For example, what is represented in Figure 1B (and Figure 5A, bottom row) and in Figure 1D? (besides, the unit of measure for the scale bar in all the figures is wrong -is it a problem of the version received for the revision?- please, check and modify if necessary). Figure 1C is puzzling to me: where was histomorphometry performed? if the trabecular number and thickness decrease in the presence of LPS, one would expect trabecular spacing increases. So please check the data, modify if necessary; if the datum is confirmed, please explain this counterintuitive observation. A similar observation applies to Fig5B, Tb.Sp. In summary, all the figure legends require revision for a common defect. Specific observations are presented, too.

Methods:

please specify whether C57BL/6J or C57BL/6N were used in this study (as the authors may know, this is not a trivial detail)

page 7 lines 241-245: please better clarify how the in vivo treatment with ibudilast was performed: administration on three consecutive days starting when? interfemoral should be intrafemoral; mg/kg.day should be mg/kg/day.

Discussion, subsection 3.2: actually, the detailed description of the molecular mechanisms involved, based on the literature, is not paralleled by investigation of these pathways/molecules in the present work; which would improve the manuscript. On this basis the authors should also perform TRAP and ALP staining on the bone tissue, and corresponding quantization, to further support their conclusions.

Minor:

Page 3: "LPS also directly and significantly increased the activity of osteoclast-associated molecules, including cathepsin K and tartrate-resistant acid phosphatase ". Actually, the authors did not evaluate enzymatic activities so please modify this sentence. On the other hand, they might assess the resorption activity of the differentiated osteoclasts. 

OS is an atypical acronym to indicate osteogenic induction medium. Would it be possible to change it and use OIM?

In the graph in Figure 2E, OIM should be indicated before LPS, since it is a control condition.

Figure 4A: Ca2+ stain should be better indicated as Alizarin Red staining.

Page 2, lines 48-50: "The systemically circulating endotoxin derived from antibiotic-killed bacteria, minor infections, intestinal flora, or dental procedures may bind to wear particles". This sentence requires rephrasing. Line 80, osteocalcin contraction should be osteocalcin concentration.

Page 4, lines 127-129: "Masson’s trichrome staining revealed bone bridge formation on day 14 after the bone was defected (Figure 5C). In the ibudilast-treatment group, a dense bone bridge appeared after introducing femoral defects." These sentences require rephrasing, with particular attention to the parts in bold.

Page 6, line 191:"The femoral intramedullary canal was intrafemorally injected" please avoid repeating the concept; "The femoral medullary canal was injected" will be clear enough.

lines 193-194:"Finally, the painkiller buprenorphine (0.2 mg/kg) was administered subcutaneously every 24 h as an analgesic throughout the experimental duration" please avoid repeating the concept; "the painkiller buprenorphine (0.2 mg/kg) was administered subcutaneously every 24 h throughout the experimental duration"" will be clear enough.

line 197: "Serum Osteocalcin and CTX1 assay" better measurement.

line 203, microCT: "For the detailed procedure, please refer to our previously published studies". Despite this, please briefly describe how and where it was performed.

line 207: "Femur samples were incubated in a rapid decalcifier solution" probably should be decalcifying; please, modify and also detail what it is.

Reviewer 2 Report

This is a succinct, well-written article related to mechanisms of LPS and its inhibition on bone regeneration. The findings are perhaps not surprising due to the authors' other related publications, but the approach and results are significant. Some issues must be addressed prior to publication, in particular, the µCT.

Major Comments:

All Figures need to be reformatted so they are visible. The µCT and cell images are too small to properly interpret in the formatted article.

Fig 1B: The different orientations and defect sites at d14 suggest heterogeneity in the surgical approach. This is concerning if these images are representative.

Fig 1C: Bone Volume (BV) and/or mineral content should be reported, or if not, a reason should be provided. As stated in the text, trabecular analysis is computationally difficult and not appropriate for this kind of cortical defect, especially when non-healing is observed.

Fig 5A: Is the majority of bone not forming in the defect, but outside the rest of the cortex?? It doesn't look like there is any boney bridge formation here, just periosteal activation outside of the defect. Perhaps this just a problem with figure size/clarity, but it puts the relevance of the surgical model into question.

Mouse/experimental n-numbers are not reported.

Minor Comments:

Lines 191, 204: Briefly describe so that readers aren't required to read two additional articles to understand the basic methods. For example, your µCT resolution isn't stated in the article.

Fig 1D: As stated, histology is too small, but also, the cortex is not visible in d14 sham?

Line 243: What is the vehicle for ibudilast injection?
